# What are the sources of stress and distress for general practitioners working in England? A qualitative study

Ruth Riley,[1] Johanna Spiers,[2] Marta Buszewicz,[3] Anna Kathryn Taylor,[4] Gail Thornton,[2] Carolyn Anne Chew-Graham[5]

[1]Institute of Applied Health Research, University of Birmingham, Birmingham, UK
[2]Bristol Medical School, University of Bristol, Bristol, UK
[3]Research Department of Primary Care and Population Health, University College London, London, UK
[4]Faculty of Health Sciences, University of Bristol, Bristol, UK
[5]Primary Care and Health Sciences, Research Institute, Keele University, Keele, UK

**Correspondence to**
Dr Ruth Riley;
r.riley@bham.ac.uk

## ABSTRACT

**Objectives** This paper reports the sources of stress and distress experienced by general practitioners (GP) as part of a wider study exploring the barriers and facilitators to help-seeking for mental illness and burnout among this medical population.

**Design** Qualitative study using in-depth interviews with 47 GP participants. The interviews were audio-recorded, transcribed, anonymised and imported into NVivo V.11 to facilitate data management. Data were analysed using a thematic analysis employing the constant comparative method.

**Setting** England.

**Participants** A purposive sample of GP participants who self-identified as: (1) currently living with mental distress, (2) returning to work following treatment, (3) off sick or retired early as a result of mental distress or (4) without experience of mental distress. Interviews were conducted face-to-face or over the telephone.

**Results** The key sources of stress/distress related to: (1) emotion work—the work invested and required in managing and responding to the psychosocial component of GPs' work, and dealing with abusive or confrontational patients; (2) practice culture—practice dynamics and collegial conflict, bullying, isolation and lack of support; (3) work role and demands—fear of making mistakes, complaints and inquests, revalidation, appraisal, inspections and financial worries.

**Conclusion** In addition to addressing escalating workloads through the provision of increased resources, addressing unhealthy practice cultures is paramount. Collegial support, a willingness to talk about vulnerability and illness, and having open channels of communication enable GPs to feel less isolated and better able to cope with the emotional and clinical demands of their work. Doctors, including GPs, are not invulnerable to the clinical and emotional demands of their work nor the effects of divisive work cultures—culture change and access to informal and formal support is therefore crucial in enabling GPs to do their job effectively and to stay well.

## INTRODUCTION

Compared with the general population, doctors, including general practitioners (GP), continue to experience high

### Strengths and limitations of this study

► Few studies employing qualitative methods, using in-depth interviews, have been used to examine this topic—this study carried out 47 interviews with general practitioners (GP) from across England and contributes to a growing body of research illuminating and examining the causes and impact of chronic stress and distress among GPs.

► Participants were self-selecting, which may be perceived as a limitation; however, the sampling of participants ensured that the sample was varied in terms of age, gender, number of years as practising GPs, level of seniority/employment status in the practice, and geographical location.

► Due to time constraints, the researchers were unable to employ respondent validation; however, the second coders included academic GPs and team members with lived experience of mental illness which afforded some checks and balances to the validity of the analytic process, interpretation of data and generalisability of the research findings.

► While recruiting individuals with experience of mental illness and burnout, we also included the perspectives of GPs who had no disclosed history of mental illness, which enables the data to be more widely generalisable to the GP population.

► The multidisciplinary research team independently analysed a subset of transcripts in order to contribute to the generation and refinement of codes to maximise rigour; emergent themes were subsequently discussed by the whole team to ensure credibility and confirmability.

levels of workplace stress and burnout,[1–4] with some evidence suggesting higher rates of mental illness.[5–9] The pressures facing GPs have been compounded by escalating bureaucracy, increased patient demand (evidenced by a 16% increase in workload over the past 7 years[10]), as well as workforce shortages and a reduction in resources.[11 12] Currently, 90% of patient consultations take place in general practice, yet the allocated budget accounts for

only 8% of the National Health Service (NHS) total.[13] [14] Consequently, the pressure on primary care is currently at its highest ever, and is predicted to increase in the future.[13] It is argued that these pressures have contributed to low job satisfaction and low morale among staff, as well as stress, burnout and early retirement,[15–17] which further compounds workforce shortages.[18] In a recent survey,[19] 20% of GPs indicated that the likelihood of leaving their job was 'high', while reported levels of stress were higher than previous years. In addition to the personal cost to individuals in terms of mental distress, the financial cost of losing doctors to burnout, early retirement and reduced clinical hours already impacts adversely on the sustainability of adequate patient services.[15] [16]

Additionally, a substantive body of evidence has identified a clear link between the experiences and well-being of NHS staff, including doctors, and the experiences of patients and the quality of care they receive, as indicated by health outcomes and patient satisfaction and quality of care ratings.[20–23] Patient reported experiences are better when staff are satisfied and feel more supported by the organisation and their supervisors.[23] It is important to understand the sources of the increased levels of stress and distress, in order to address them more effectively. This paper reports the sources of stress and distress experienced by GPs as part of a wider study exploring the barriers and facilitators to help-seeking for mental illness and burnout among this medical population.

## METHODS

This was a multicentre qualitative study, employing in-depth interviews with 47 GPs in England. Information about the study was advertised through professional publications such as Pulse, social media, and national and local GP networks (such as Local Medical Committees and Clinical Commissioning Groups) in Bristol, Manchester and London. A subsample (n=12) was recruited through a specialist treatment service. Interested GPs were sent a participant information sheet and informed consent was taken from those willing to take part. GPs wishing to take part were asked to self-select into the following groups: (1) living with anxiety, depression, stress and/or burnout; (2) returning to work following treatment; (3) off sick or retired early due to illness; (4) no mental illness. Participating GPs were reimbursed with £80 to recognise the time for their participation. Interested GPs were purposively sampled to represent as even a spread as possible across these four groups, although the largest number of participants was in group 1. We intended to purposively sample approximately 10 participants per group. However, the majority of GPs who contacted us self-selected into group 1, and due to the emergent rich data continued recruitment to this group and further exploration of emerging themes was justified in meeting the study's aims and objectives. We endeavoured to recruit more participants into groups 2 and 4 using targeted publicity information, but because of time constraints

**Box   Interview topic guide**

Introduction and background
► Describe average working day in practice (hours, surgery, home visits) and any additional responsibilities.

Current well-being
► Describe average working day in practice (hours, surgery, home visits) and any additional responsibilities.
► Explore current well-being, feelings about work, levels of stress, work-life balance.
► Explore causes of stress/distress (workload, hours, admin, clinical caseloads, organisational issues, lack of support, personal issues, pre-existing mental health symptoms).
► Explore reasons for early retirement/sickness (if relevant).

Managing stress
► Explore how general practitioners (GP) manage their workload/stress in their day-to-day work life, what they do to relax, to look after themselves (self-care strategies: supportive relationships, sport, exercise relaxation techniques).
► Explore relationship with colleagues and whether/how/if concerns are raised, how they are responded.
► Explore if given informal/formal supervision or mentor (1:1 or group) and experience/value of group.
► Explore thoughts/feelings about seeking help, barriers to seeking help (stigma/shame, fears about confidentiality, uncertainty of where to go).

those groups remained marginally under-recruited. In the event, many GPs who identified as living with no stress reported as having had experiences of stress and distress at some juncture in their career.

More female GPs contacted the study team expressing an interest in participating, and therefore the disparity in numbers reflects this. The iterative process of recruitment, sampling and analysis ensured that emerging concepts and themes could be tested out among participants with different characteristics (eg, partners vs locum GPs). Further GPs who expressed an interest in participation were politely thanked, given an explanation about the high rates of interest in the study and told that their participation was not required.

Face-to-face (either at the participant's home, or at work) or telephone interviews lasting between 27 and 126 min (mean=69 min) were conducted between April and September 2016. The recorded in-depth interviews were conducted by two authors (JS, RR), both social scientists and qualitative experts, using a flexible topic guide (see box). This was informed by the existing literature, input from GPs on the study team and patient and public involvement (PPI) consultation exercises conducted with GP networks prior to obtaining funding. The interviews were audio-recorded, transcribed, anonymised and imported into NVivo V.11 to facilitate data management. Analysis and data collection were conducted iteratively until data saturation was reached and no new themes were arising from the data.[24] A thematic analysis was conducted, involving a process of constant comparison between cases.[25] Analysis commenced with JS generating an initial coding framework, grounded in the data, which

was added to and refined, with material regrouped and recoded as new data were gathered. Codes were gradually built into broader categories through comparison across transcripts and higher level recurring themes were developed. Reflexivity was employed throughout the research. Both interviewers were experienced qualitative researchers who both reflected on and discussed the impact of the data on their cognitive and emotional sensing throughout the study.[26] Both researchers also discussed and made explicit how their epistemological (JS with a background in psychology and RR in medical sociology) and experiential backgrounds may have oriented the data collection and analytic process.

The multidisciplinary research team independently analysed a subset of transcripts in order to contribute to the generation and refinement of codes to maximise rigour. Emergent themes were discussed by the whole team to ensure credibility and confirmability.

## RESULTS

Forty-seven interviews were conducted with participants. The demographic and practice characteristics of participants are included in table 1.

Analysis of the interview transcripts and field notes identified three main themes, with corresponding subthemes relating to sources of stress/distress:

► *Emotion work*—the work invested and required in managing and responding to the psychosocial component of GPs' work and dealing with abusive or confrontational patients.
► *Practice culture*—practice dynamics and collegial conflict, bullying, isolation and lack of support.
► *Work role and demands*—fear of making mistakes, complaints and inquests, revalidation, appraisal, inspections and financial worries.

### Emotion work

This theme refers to the emotion work invested and required in managing and responding to the emotional content of consultations with patients. In particular, the work required in managing confrontational patients and in treating and responding to the many psychosocial components of a GP's work and its subsequent impact on clinicians, as these participants indicate:

I think when you're dealing with really difficult problems, you're dealing with lots of sadness, you're dealing with loads of stuff that you can't change, and people bring in and they park with you their problems and their sadnesses, and they feel better for that and you feel worse. (P20, male partner)

It's like they're [patients] hanging tiny weights on you. (P21, female salaried and locum)

However, it is important to remember that while face-to-face consultations with patients can be emotionally demanding, many participants indicated that patient

| Table 1 Participant demographics | |
|---|---|
| **GP characteristics** | **n=47** |
| Sex (female) | 33 |
| Age (years) | |
| 20–29 | 1 |
| 30–39 | 12 |
| 40–49 | 20 |
| 50–59 | 14 |
| Group | |
| 1 | 19 |
| 2 | 9 |
| 3 | 11 |
| 4 | 8 |
| Years since qualified | |
| <10 | 19 |
| No of sessions contracted per week | |
| <5 sessions (mean actual hours worked) | 12 (15) |
| >5 sessions (mean actual hours worked) | 32 (38) |
| Fully retired | 3 |
| Mean size of practice | 12 624 |
| Range | 3600–37 000 |
| Employment status | |
| Partner | 17 |
| Salaried | 11 |
| Locum | 5 |
| Registrar | 4 |
| Retired | 3 |
| Sick leave | 5 |
| More than one role | 2 |

GP, general practitioner.

contact was also the most enjoyable and satisfying aspect of their work, as these participants highlight:

My job satisfaction was patients coming in and going out having got their needs attended to and happy. (P49, female retired)

I feel very conflicted, because I love general practice. I've always wanted to be a GP. I love my patients. And on a day when I've got time to spend with them, it's a brilliant job. (P17, female partner)

A related source of stress/distress was the emotion work required in managing physically and verbally abusive or confrontational patients—another common theme reported among participants:

You know, it would be very much around, you know, patients who'd been quite aggressive, being quite rude, or having a particularly busy night. (P12, female partner)

I've had situations where I've felt say physically

threatened and had to ask someone to leave. (P6, male salaried)

## Practice culture
### Practice dynamics and collegial conflict

The stress and anxiety associated with the responsibilities related to staffing issues, practice dynamics and collegial conflict was a key source of stress among GP participants, as these participants indicate:

> Some of the stress, in general practice as distinct from other branches of medicine, comes from the internal politics. And there have been some very stressful times with partners. (P1, female partner)

> And then I became more unhappy with the practice. Now that's not necessarily the patients and the work of the practice. It's the difficulty I had in relationships with partners. (P51, male retired)

Stresses related to power politics within the practice, having decisions undermined or overturned, or as a result of a lack of input in decision-making were also described by some:

> And increasingly I felt that my view, decisions I think we've made, have been overturned. By the meeting. And so I felt increasingly I had less and less say in the practice and that my views weren't taken into account. (P46, female partner)

> I just seemed to be getting over-ruled by the senior partner. (P47, female on sick leave)

### Bullying

The experience of being bullied by colleagues and/or partners was a recurring theme and a key source of distress for a significant minority of participants in this study. This bullying contributed to or exacerbated ongoing chronic stress among participants, and also contributed to high staff turnover in some practices:

> I thought, 'This is like an abusive relationship, where I'm unwell and I'm being shouted at at work. I hate coming into work. I'm crying a lot with, you know, finding it so stressful.' (P31, female locum)

> And it was just one (pause) there was one particular individual who made my life very difficult, who really I found very difficult through the whole time I was in the partnership. (P22, male salaried)

These participants highlighted that colleagues, including partners, could be complicit in perpetuating abuse among staff, through their lack of support, ambivalence or by failing to intervene:

> Yeah, and they had a huge turnover of GPs […] There was one GP who was a bit of a bully and the rest were just unsupportive. And the patients were fed up because they had had so many changes of GPs. (P37, female off sick)

### Isolation and lack of support

Feeling isolated and unsupported in their work was a dominant theme among the majority of the 47 GP participants, and contributed to participants' existing stress and distress. The loneliness and isolation was also compounded by the burden of responsibility and increasing workload, which left GPs with little time to connect with their colleagues. Some participants highlighted that a sense of isolation was often heightened in general practice, as most of their work was done on an individual basis:

> You don't really leave your room or talk to many other people. (P26, female salaried)

> And I think that's one of the issues that general practice particularly faces, in that you're actually quite isolated. (P29, female partner)

The following participant highlights that this may be a particular occupational hazard for locums, who can feel disconnected from their colleagues:

> [Being a locum is] quite isolating in that you could go to a practice for the first time, never meet any of the other doctors, hardly see a nurse. You're just told, 'Right, here's your room. Get on with it,' and you just see a patient every ten minutes. And, yeah, I find that quite difficult. (P5, female salaried and locum)

GPs' sense of working alone was exacerbated by the increase in workload, resulting in fewer opportunities for a break and time to talk to colleagues:

> But as the workload increased, the coffee time became less (pause) important in everyone's day, because they would catch up with their paperwork or phone calls or extras […] Over those years we, I think we all became more isolated and separate from one another. (P35, female retired)

The lack of support and acknowledgement among other colleagues or GP partners for those who were experiencing mental illness was also observed and experienced by many participants:

> One of my colleagues went off with depression […] and then died by suicide, which was really devastating to the practice. But it was a bit like, 'Well, we've just got to carry on. We've got this job to do,' and there was no time to (pause) even really acknowledge it, you know, beyond having the usual things like funerals and stuff. So it was almost swept under the carpet like, 'Well, you know, let's just carry on. Let's not address what perhaps might have contributed to it at work.' (P31, female locum)

The following participant highlights the contrast between supportive practices and those which appeared to engender a 'survival of the fittest' culture:

> There may be practices that are really supportive and really want to know these things and want to help

people, (laughs) but that's certainly not the case in ours. Every man for himself, I think. (P9, male partner)

In contrast, some GP participants clearly benefited from being part of a team and working alongside colleagues who have invested in cultivating a supportive culture over time:

I work in a fantastic team. I love all my partners. I love the team I work with. And that's something that we have created over, over that, all those years. You know, it's not just something I parachuted into; it's been carefully created by me and my partners over that time. (P19, male partner)

The following quote highlights the importance of support and collegiality as a protective factor for good mental health, maintaining morale and feeling less isolated in dealing with the challenges of being a GP:

The ones that are left of us at the moment are being very supportive of each other and making sure that we're not getting too bogged down with work each and that we are OK. And we keep checking on each other and, you know, 'Are you alright?' And we've had a few little social things to try and get everyone's morale up and support each other and things like that. (P27, female partner)

### Work role and demands

The roles and demands intrinsic to the work of a GP were a source of significant stress for participants. They frequently reported stress related to workload and long hours; however, we have omitted these findings since they have been widely reported previously (see the Discussion section). Other work-related sources of stress among participants included a fear of making mistakes (compounded by an increased workload), complaints and inquests, as well as the additional work associated with revalidation and appraisal, and managing finances on reduced budgets.

#### Fear of making mistakes

A significant source of palpable anxiety and distress for many GP participants was the fear of making a clinical mistake and the resulting repercussions. Many participants were concerned that the escalating workload directly increased the risk of making a mistake:

The fear of making mistakes is—huge. When you see really good colleagues who, because you're dealing with such an overwhelming volume of work, making hundreds and hundreds and hundreds of clinical decisions a day—they're not all gonna be right. (P24, female partner)

#### Managing complaints

The stress of managing complaints from patients was also a common theme:

A serious complaint is a failure, isn't it? You've done something badly wrong probably, or someone thinks you've done something badly wrong. And they can be very difficult times emotionally for doctors. And um often they arrive at the times when you're most vulnerable, so quite a few people are very badly affected by complaints. (P36, male partner)

#### Revalidation, appraisal and inspections

The additional workload and stress associated with undertaking revalidation, appraisal and care quality commission (CQC) inspections was a dominant theme among the participants, some of whom found the process to be punitive, unhelpful and at times overwhelming:

…a lot of younger doctors do get very stressed about appraisals actually. I mean older doctors do as well, because of all the information collecting and it just feels like, 'Oh bloody hell, I'm doing the job. Can't you see I'm doing it alright? Why do I have to prove I'm doing it alright?' (P2, female partner)

A lot of the additional work and tick-boxing and bureaucracy we're being asked to do could be done away with. The CQC thing has kind of exploded from nowhere. We're being charged thousands of pounds (pause) to provide evidence of X, Y and Z, and there is no basis that this proves that one GP surgery is better than another, it's just an enforced grading system. (P13, male partner)

#### Financial risks for partners

Being a GP partner usually brings with it a financial burden. There was concern among participants that general practices are being expected to do more on fewer resources, which also has financial implications for individuals:

And it's also very worrying because, financially, you know, as a partner in the practice, if it goes under and we're at the stage where, we are very, very close to it going under (pause) I will be personally responsible for hundreds and hundreds of thousands of pounds of debt. Which means that we will lose our house. (P24, female partner)

### DISCUSSION

These findings indicate that the sources of stress and distress for GPs are varied and relate to the occupational pressures of working as a GP, including the emotional pressures involved in managing the psychosocial elements of everyday consultations, as well as having to manage some abusive, confrontational patients and dealing with complaints. Stress and distress were also associated with the practice environment, including dysfunctional relationships at work, collegial conflict, bullying and the lack of support from colleagues, which further compounded the sense of isolation often experienced in their work.

Further key sources of stress were long hours, time pressures, the fear of making mistakes and inquests, as well as the appraisal and revalidation processes. Participants who were partners also reported tensions relating to their financial responsibilities and being expected to achieve more on reduced budgets while managing personal debt accrued by buying into the practice.

Many of these stressors were interlinked and cumulatively contributed to and/or exacerbated existing distress. Participants frequently worked with little support or supervision, often feeling isolated within their practice. Crucially, those GP participants who felt more supported in their practices, as a result of greater collegial support, colleagues' willingness to talk about vulnerability and illness and having open channels of communication within the practice, reported feeling less isolated, more resilient and better able to cope with the emotional and clinical demands of their work.

The existing literature has identified increasing workloads, time pressures, long hours and bureaucratic demands as key causes of work-related stress/distress among GPs.[5 11–13 27 28] Suggestions for addressing stresses intrinsic to the work of a GP (ie, workload, bureaucracy) have also been identified.[12 27 28] Participants in this study also highlighted sources of stress associated with the fear of making mistakes, inspections, complaints and inquests, as well as the pressures associated with the revalidation and the appraisal process—the latter regarded as unhelpful, time consuming and of little value.[29] The emotional component of work in general practice and its impact on GP well-being is supported by previous evidence highlighting the impact of the emotional demands of working with patients and exposure to suffering.[5]

This study and recent findings[12] suggest that unhealthy practice cultures (which may be characterised by bullying or the absence of collegial support and the opportunity to discuss the emotional and clinical demands of GPs' work) are a key source of stress and distress for many GPs. As highlighted by the participants, working in an unsupportive cultural climate can cause distress and add to a burgeoning sense of isolation for many individuals, while the attitudes of some senior colleagues appeared to create or perpetuate a culture of distancing and disengagement with one's feelings and those of colleagues. Such a culture sets an unhealthy and unhelpful precedent for how feelings are expressed and managed within the work setting, which can exacerbate existing feelings of isolation and difficulties in seeking effective support, particularly for GPs in distress and those living with mental illness.

These findings have important implications for policy and practice, namely that providing a safe space for GPs to process the emotional and clinical content of their work and the potential stressors related to the organisational culture (eg, bullying in the workplace) and relationships at work (eg, collegial conflict) is imperative. This echoes findings from a recent systematic review on interventions to reduce burnout among doctors, including GPs, which found that improving communication between team members, cultivating teamwork, increasing job control and giving doctors permission to acknowledge and manage stress have proven effective in militating against burnout.[30] GPs are expected to provide their patients with the space, opportunity and permission for them to communicate their concerns and feelings within the consultation, thus enabling individuals to voice and share their experiences and be heard.[31] However, the same is not always afforded to GPs.

In terms of prevention and provision, NHS England have committed to increasing the retention, return and recruitment of GPs and investing more resources in general practice aiming to reduce the workload, as well as providing a specialist England-wide occupational health service for GPs.[32] This new GP occupational health service is also available to support individuals who are affected by toxic work cultures such as the bullying, collegial conflict and practice dynamics highlighted in this research. The ability to respond effectively to the emotional demands and anxiety often expressed by patients in the consultation[33] without feeling overwhelmed needs to be addressed in GP training as well as in ongoing support and supervision. Crucially, collegial support is a protective factor for good mental health—support from mentors, supervisors and colleagues is associated with resilience and reduced sickness.[34] Balint groups or similarly structured group work or supervision continue to be employed in general practice and are valued by GPs,[35] yet are optional. Individual or group supervision aims to provide a safe and supportive space where staff can openly discuss the pressures and emotional challenges of their work and may, as previous evidence suggests, provide GPs with the support they need while offering protection against compassion fatigue and burnout.[36] Their wider use may need to be reconsidered. Tackling the culture of invulnerability early on in medical training is also key. Schwartz Centre Rounds, for instance, are currently being piloted in medical schools with early evidence supporting their value.[37] First 5, buddy systems and access to regular supervision or mentorship can also offer support and a reflective space—talking, sharing and having one's feelings normalised, understood and validated are critical in maintaining good mental health.

## CONCLUSION

This study highlights that the sources of stress and distress cannot solely be attributed to increases in workload and occupational stress linked to the work role demands of being a GP. Sources of stress and distress are also linked to unhealthy practice cultures, which may be characterised by bullying or the absence of collegial support and the opportunity to discuss the emotional and clinical demands of GPs' work. Such a workplace climate can cause and add to stress and distress, and leave GPs feeling isolated. Promoting compassionate and supportive work cultures is therefore imperative in addressing the dynamic interplay between the personal,

professional and organisational sources of stress and distress for GPs.

**Acknowledgements** The authors thank all volunteer participants who kindly gave their time to share their experiences of working in English general practice; and collaborators Dr Clare Gerada and Dr Chris Manning for endorsing this study and who provided valuable comments on the funding application and who assisted with recruitment.

**Contributors** RR, JS, MB and CCG: substantial contributions to conception and design, acquisition of data, or analysis and interpretation of data; drafting the article or revising it critically for important intellectual content; and final approval of the version to be published. AT and GT: analysis and interpretation of data; drafting the article or revising it critically for important intellectual content; and final approval of the version to be published.

**Funding** The study was funded by the NIHR School for Primary Care Research.

**Disclaimer** The views and opinions expressed therein are those of the authors and do not necessarily reflect those of the NIHR School for Primary Care Research, NIHR, NHS or the Department of Health.

**Competing interests** None declared.

**Ethics approval** Ethical approval was granted by the South West-Frenchay Research Ethics Committee (reference number: 15/SW/0350).

**Provenance and peer review** Not commissioned; externally peer reviewed.

**Data sharing statement** This study has not received ethical approval to share confidential data with any third party other than the study research team.

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
