## [Reviewer comments · BMJ Open]

ARTICLE DETAILS

TITLE (PROVISIONAL)	What are the Sources of Stress and Distress for General Practitioners Working in England? A Qualitative Study
AUTHORS	Riley, Ruth; Spiers, Johanna; Buszewicz, Marta; Taylor, Anna; Thornton, Gail; Chew-Graham, Carolyn

VERSION 1 - REVIEW

REVIEWER	Marylou Murray Centre for Public Health Institute of Clinical Sciences, Block B Queens University Belfast Royal Victoria Hospital Belfast BT12 6BA
REVIEW RETURNED	11-May-2017

GENERAL COMMENTS	The authors are to be commended for this timely, informative contribution to GP health and wellbeing research. Here are some minor points they may wish to consider. Abstract: Page 2, Lines 48-53 Do the study objectives include insights into stress management techniques/coping strategies (as suggested by the Interview topic guide) in addition to sources of stress and distress? Strengths and limitations: Page 3 line 18...are there some words missing e.g. In 'addition to recruiting?' Introduction: Page 3 lines 32-34 - the references cited do not adequately support the claim of higher mental illness prevalence in doctors than in the general population. In view of the dearth of relevant high quality evidence this statement would benefit from moderation or qualification. Page 3 Lines 50-54 - the references cited pertain in the main to secondary care. There is contrary evidence in one of the paper's own references (Orton et al 2012) Other references throughout the manuscript are appropriate and up to date. Methods: 47 indepth interviews is a considerable body of work. More detailed justification of the sampling strategy would be helpful. There appears to be maximum variation within the sampling however there is considerable disparity in group sizes. The authors report that data
--

	collection and analysis were conducted iteratively...how does this relate to recruitment? Is data saturation relevant? Was there any respondent validation? Report of reflexivity would provide additional procedural rigour. Results: Clearly presented with quotations that make results understandable. Discussion: Key findings are clearly presented. Interpretation is innovative and credible. Conclusion: Clear synthesis of the study with useful recommendations.
--	---

REVIEWER	Daniel Jones Hull York Medical School UK
REVIEW RETURNED	24-May-2017

GENERAL COMMENTS	I enjoyed reading the manuscript and found it very well written and relevant. As a locum GP many of the finds strike a chord with many of my experiences. My only thoughts are with what we do next, especially regarding the interesting findings of practice culture. I can't help but think that mentorship and discussion groups will be ineffective in resolving issues such as bullying, collegial conflict and practice dynamics. I wonder if the authors could think of any other possible solutions or help for GP colleagues in this difficult situation.
---

REVIEWER	Martin Roland University of Cambridge
REVIEW RETURNED	29-May-2017

GENERAL COMMENTS	This is an excellent paper which addresses an important issue. It's well written and the data presented (quotations) support the analytic themes and conclusions drawn. There is an issue about the self-selecting nature of the sample, but I don't think this is a concern and it's adequately discussed by the authors.
---

VERSION 1 – AUTHOR RESPONSE

Reviewer 1:

The authors are to be commended for this timely, informative contribution to GP health and wellbeing research.

Author's Response: Thank you for this positive comment

Do the study objectives include insights into stress management techniques/coping strategies (as suggested by the Interview topic guide) in addition to sources of stress and distress?

Author's Response: Yes, we explored participants' coping strategies and these have been reported in greater depth in a paper focusing on the barriers and facilitators to help-seeking, submitted elsewhere (BJGP).

Strengths and limitations:

Page 3 line 18...are there some words missing e.g. In 'addition to' recruiting?

Author's Response: We have amended this sentence on page 3

Introduction:

Page 3 lines 32-34 - the references cited do not adequately support the claim of higher mental illness prevalence in doctors than in the general population. In view of the dearth of relevant high quality evidence this statement would benefit from moderation or qualification.

Author's Response: We have moderated and qualified this statement concerning rates of mental illness on page 3.

Page 3 Lines 50-54 - the references cited pertain in the main to secondary care. There is contrary evidence in one of the paper's own references (Orton et al 2012). Other references throughout the manuscript are appropriate and up to date.

Author's Response: We have moderated and qualified this statement concerning rates of mental illness on page 3.

Methods:

47 in-depth interviews is a considerable body of work. More detailed justification of the sampling strategy would be helpful. There appears to be maximum variation within the sampling however there is considerable disparity in group sizes. The authors report that data collection and analysis were conducted iteratively...how does this relate to recruitment?

Author's Response: We agree, there is disparity across the groups in terms of numbers and have provided a more detailed explanation in the text on page 4:

We intended to purposively sample approximately 10 participants per group. However, the majority of GPs who contacted us, self-selected into group one and due to the emergent rich data, further recruitment and exploration of emerging themes was justified in meeting the study's aims and objectives. We endeavored to recruit more participants into groups two and four using targeted publicity information but due to time constraints, those groups remained marginally under-recruited. In the event, many GPs who identified as living with no stress reported as having had experiences of stress and distress at some juncture in their career.

More female GPs contacted the study team expressing an interest in participating, and therefore the disparity in numbers reflects this. The iterative process of recruitment, sampling and analysis ensured that emerging concepts and themes could be tested out amongst participants with different characteristics (i.e. partners vs locums GPs).

Is data saturation relevant?

Author's Response: Thank you for this comment. We have included an explanatory statement in relation to data saturation on page 4 with an additional reference to support this: Data collection and analysis were conducted in parallel and interviews continued until data saturation was reached and no new themes were arising from the data (Sandelowski, 1995)

Report of reflexivity would provide additional procedural rigour.

Author's Response: We have included the following statement to provide additional procedural rigour: Both interviewers are experienced qualitative researchers and both reflected on and discussed the impact of the data on their cognitive and emotional sensing throughout the study. Both researchers also discussed and made explicit how their epistemological (JS with a background in psychology and RR in medical sociology) and experiential backgrounds may have oriented the data collection and analytic process. See page 4/5

Reviewer 2

I enjoyed reading the manuscript and found it very well written and relevant. As a locum GP many of the finds strike a chord with many of my experiences.

Author's Response: Thank you for this positive comment

My only thoughts are with what we do next, especially regarding the interesting findings of practice culture. I can't help but think that mentorship and discussion groups will be ineffective in resolving issues such as bullying, collegial conflict and practice dynamics. I wonder if the authors could think of any other possible solutions or help for GP colleagues in this difficult situation.

Author's Response: The authors acknowledge the challenges in addressing negative organisational cultures. The new occupational health service for GPs is available to support individuals facing such difficulties. We have included a sentence in the discussion, on page 11: This new occupational health service for GPs is also available to support individuals who are affected by toxic work cultures such as the bullying, collegial conflict and practice dynamics highlighted in this research.

Reviewer 3

This is an excellent paper which addresses an important issue. It's well written and the data presented (quotations) support the analytic themes and conclusions drawn. There is an issue about the self-selecting nature of the sample, but I don't think this is a concern and it's adequately discussed by the authors.

Author's Response: Thank you for this positive comment. We feel that the self-selecting nature of the sample is a strength of the study, and that we have reached a group of GPs whose voices might not otherwise be heard.

VERSION 2 – REVIEW

REVIEWER	Marylou Murray Centre for Public Health Queen's University Belfast
REVIEW RETURNED	04-Jul-2017
GENERAL COMMENTS	This revised manuscript comprehensively addresses all issues raised at initial review.